# Aspiletrein A Induces Apoptosis Cell Death via Increasing Reactive Oxygen Species Generation and AMPK Activation in Non-Small-Cell Lung Cancer Cells

**DOI:** 10.3390/ijms23169258

**Published:** 2022-08-17

**Authors:** Wasita Witayateeraporn, Hien Minh Nguyen, Duc Viet Ho, Hoai Thi Nguyen, Pithi Chanvorachote, Chanida Vinayanuwattikun, Varisa Pongrakhananon

**Affiliations:** 1Department of Pharmacology and Physiology, Faculty of Pharmaceutical Sciences, Chulalongkorn University, Bangkok 10330, Thailand; 2Faculty of Pharmacy, Ton Duc Thang University, Ho Chi Minh City 700000, Vietnam; 3Faculty of Pharmacy, Hue University of Medicine and Pharmacy, Hue City 49000, Vietnam; 4Division of Medical Oncology, Department of Medicine, Faculty of Medicine, Chulalongkorn University, Bangkok 10330, Thailand; 5Preclinical Toxicity and Efficacy Assessment of Medicines and Chemicals Research Unit, Chulalongkorn University, Bangkok 10330, Thailand

**Keywords:** aspiletrein A, AMP-activated protein kinase, apoptosis, non-small-cell lung cancer cells, reactive oxygen species

## Abstract

Lung cancer remains a leading cause of death in cancer patients, and deregulation of apoptosis is a serious concern in clinical practice, even though therapeutic intervention has been greatly improved. Plants are a versatile source of biologically active compounds for anticancer drug discovery, and aspiletrein A (AA) is a steroidal saponin isolated from *Aspidistra letreae* that has a potent cytotoxic effect on various cancer cell lines. In this study, we investigated and determined the underlying molecular mechanism by which AA induces apoptosis. AA strongly induced apoptosis in NSCLC cells by mediating ROS generation and thereby activating AMP-activated protein kinase (AMPK) signaling. Consequently, downstream signaling and levels of phosphorylated mTOR and Bcl-2 were significantly decreased. Pretreatment with either an antioxidant, N-acetylcysteine, or an AMPK inhibitor, compound C, could reverse the apoptosis-inducing effect and counteract the effect of AA on the AMPK signaling pathway. Decreased levels of Bcl-2 were due to AA-mediating Bcl-2 degradation via a ROS/AMPK/mTOR axis-dependent proteasomal mechanism. Consistently, the apoptotic-inducing effect of AA was also observed in patient-derived malignant lung cancer cells, and it suppressed an in vitro 3D-tumorigenesis. This study identified the underlying mechanism of AA on lung cancer apoptosis, thereby facilitating potential research and development of this compound for further clinical implications.

## 1. Introduction

Lung cancer is an aggressive malignancy with a globally poor mortality rate [1]. The estimated deaths from lung cancer account for approximately 22% of all cancer types, and the occurrence rate is greater than that of other cancers [2]. Furthermore, the five-year survival rate is low at only 15% [3,4]. Due to unclear specific symptoms, most patients are initially diagnosed after the advanced stage has presented [5]. Non-small-cell lung cancer (NSCLC) is the major type of lung cancer accounting for 85% of all patients [6], more than half of which exhibit metastatic progression [7]. Even though current therapeutic interventions have become more advanced and diagnosis/prognosis markers have been identified [8,9,10], there is a frequent acquisition of chemoresistance [11], which limits the clinical outcome.

Apoptosis, or programmed cell death, is generally responsible for tissue maintenance and development [12]. The resistance to apoptotic cell death is commonly found in tumor pathogenesis in association with deregulation of cell death and/or prosurvival signaling [13,14,15]. AMP-activated protein kinase (AMPK) plays a crucial role in maintaining cellular energy homeostasis that is activated by energy stress [16]. Activated AMPK promotes ATP production and inhibits cellular activities with high energy consumption. In addition, AMPK acts as a tumor suppressor that induces apoptosis and inhibits cell proliferation in cancers as cancer cells exhibit a high metabolic capability in response to the energy demand for cell growth and protein synthesis [17,18]. Importantly, AMPK expression and activation are reported to be significantly lowered in lung cancer [19,20]. Additionally, enhancement of AMPK activity was able to induce apoptosis and suppress in vitro and in vivo NSCLC growth [21,22,23], indicating that AMPK is a potential therapeutic target in drug research and discovery for lung cancer treatment.

Several anticancer drugs obtained from natural sources show versatile biological effects. Aspiletrein A (AA) is a steroidal saponin isolated from *Aspidistra letreae* [24] that exhibits various pharmacological activities, such as diuretic, expectorant, and antiviral and antifungal actions [25]. The cytotoxicity of this compound against several cancer types, including breast, cervical, lung, and gastric cancer cell lines, has also been reported [25]. A recent study revealed that AA could attenuate lung cancer metastasis through suppression of the protein kinase B (Akt) signaling pathway and induced cell death by downregulation of the antiapoptotic Bcl-2 protein, whereas it has less toxicity to normal bronchial epithelial BEAS-2B cells [24]. However, the effect of AA treatment on apoptotic cell death and the interaction with AMPK have not yet been discovered. This study aimed to explore the potential apoptotic-inducing effect of AA and its underlying mechanism in NSCLC cell lines and also in primary cell cultures from patients with malignant lung cancer. We demonstrate that treatment of NSCLC cells with AA can induce apoptosis via a decrease in the level of Bcl-2 mediated by ROS generation and AMPK signaling. Thus, AA may have a role in the development of future lung cancer therapies.

## 2. Results

### 2.1. AA Mediates Cell Death via an Apoptotic Mechanism in NSCLC

We first evaluated the cytotoxic effect of AA on NSCLC in A549, H460, and H23 cell lines using the MTT assay. Treatment with AA caused a decrease in cell viability in a dose-dependent manner, with a higher potency than that of cisplatin, a first-line anticancer drug (Figure 1B). The IC_50_ values of AA were 9.60 ± 2.57, 11.43 ± 3.07, and 15.44 ± 3.29 μM in A549, H23, and H460 cells, respectively. The apoptotic mode of cell death was assessed using Hoechst 33342 staining. Treatment with AA (≥25 μM) led to a dramatic increase in apoptotic cells with nuclei fragmentation and chromatin condensation (Figure 1C). To further confirm apoptotic cell death, annexin-V/PI staining was performed. The populations of annexin-V-positive cells in the upper and lower right quadrants (Figure 1D), which indicate early- and late-apoptotic cells, respectively, were significantly elevated in response to AA treatment in all tested cells as compared with that in the control group. The population of PI-positive necrotic cells in the upper left quadrant was decreased, suggesting that AA-mediated cell death via apoptosis. In addition, the expression of caspase 3 and PARP, which are the key regulators in apoptotic signaling, was also investigated. Western blotting revealed that the ratios of cleaved caspase 3 and cleaved PARP to their total forms was remarkably upregulated in AA-treated groups as compared with those in the control (Figure 2A,B), indicating that AA-induced NSCLC apoptosis was a caspase-dependent mechanism.

### 2.2. AA Induced Apoptosis by Increasing ROS Generation

To determine whether AA enhanced ROS production and consequently induced apoptosis, we first measured the level of ROS using the DCFH_2_-DA assay. AA treatment at both 2 and 3 h greatly augmented the level of ROS in a dose-dependent manner compared with that in the control group; this increase was significantly attenuated by pretreatment with NAC, an antioxidant (Figure 3A). Furthermore, the cell viability assay revealed that the reduction in cell viability caused by AA treatment was in turn prevented by NAC pretreatment (Figure 3B). We performed Hoechst 33342 staining in the presence or absence of NAC pretreatment to examine whether ROS were associated with the apoptotic-inducing effect of AA in NSCLC. Apoptotic cells that were clearly observed with AA treatment were markedly reduced in frequency in response to NAC pretreatment (Figure 3C). Western blotting of cleaved caspase 3 and cleaved PARP also confirmed that AA treatment led to upregulation of cleaved caspase 3 and cleaved PARP, whereas pretreatment with NAC could counter this effect (Figure 3D). These results suggested that AA-induced production of ROS causes NSCLC apoptosis.

### 2.3. AA Induced Apoptosis through Activation of AMPK Signaling in a ROS-Dependent Manner

To investigate whether AA induced NSCLC apoptosis via the AMPK pathway, levels of AMPK and the downstream apoptotic-signaling targets mTOR and Bcl-2 and proapoptotic Bax were assessed using Western blotting. AA treatment activated AMPK signaling by notably upregulating the level of p-AMPK (Figure 4A) compared with those in the control group, whereas the levels of p-mTOR and antiapoptotic Bcl-2 significantly declined, especially at a dose of 50 μM in all tested cells (Figure 4A,B), while the Bax level was not altered (Appendix A). We further tested whether AA-induced ROS was involved in the activation of AMPK signaling. Cells were treated with AA in the presence or absence of NAC, and the levels of p-AMPK, p-mTOR, and Bcl-2 were examined. Western blotting revealed that an increase in p-AMPK levels activated by AA was abolished following pretreatment with NAC (Figure 4C,D). Consistent with this result, the levels of p-mTOR and Bcl-2, which were previously downregulated by AA treatment, were elevated in the presence of NAC, indicating that AA-activating AMPK signaling and apoptosis were involved with ROS generation.

To further confirm whether AA induced apoptosis through the AMPK/mTOR/Bcl-2 signaling pathway, we examined the effect of AA treatment on the cell viability of NSCLC cells in the presence or absence of CC, an AMPK inhibitor. AA treatment caused a notable decrease in cell viability, which was clearly rescued by CC pretreatment (Figure 5A). In agreement, CC pretreatment significantly inhibited AA-mediated apoptosis in all tested cells (Figure 5B). Western blotting also confirmed that the increased levels of cleaved caspase 3 and cleaved PARP, which were extensively elevated in response to AA, were markedly decreased in the presence of CC (Figure 5C). Interestingly, CC was able to attenuate the effect of AA in the activation of AMPK signaling by downregulating levels of p-AMPK and upregulating those of p-mTOR and Bcl-2 (Figure 5D). These results suggested that the apoptosis-inducing effect of AA was achieved through ROS-activated AMPK signaling and the consequent suppression of mTOR and Bcl-2 expression in NSCLC cells.

### 2.4. AA-Mediated Anti-Apoptotic Bcl-2 Proteasomal Degradation

AMPK/mTOR signaling pathway has been reported to regulate apoptosis in association with Bcl-2 [26]. As we had demonstrated that AA treatment could decrease the levels of Bcl-2 in NSCLC cells, we further examined whether a reduction of Bcl-2 was caused by AMPK/mTOR-facilitating degradation of Bcl-2. The stability of Bcl-2 was determined by the level of Bcl-2 phosphorylation (p-Bcl-2) at serine 70, which is mediated in a degradation process by the ubiquitin/proteasomal mechanism [27]. To test our hypothesis, cells were pretreated with MG132, a proteasomal inhibitor, before treatment with AA. The level of p-Bcl-2 was gradually elevated in response to AA treatment (Figure 6A), and MG132 pretreatment could rescue the downregulation of Bcl-2 mediated by AA (Figure 6B). Furthermore, the presence of either NAC or CC was able to abolish phosphorylation of Bcl-2 caused by AA treatment (Figure 6C,D), indicating that the triggering of ROS and activating of AMPK caused by AA treatment resulted in Bcl-2 downregulation via proteasomal degradation. The level of Bcl-2 mRNA was unchanged in AA-treated cells compared with that in the control, indicating that AA treatment regulated the level of Bcl-2 through a post-translational modification (Appendix A).

To confirm this finding, the half-life (T_1/2_) of Bcl-2 was assessed in the presence of AA and cycloheximide (CHX), a protein synthesis inhibitor. After treatment with CHX alone, the Bcl-2 level gradually declined with time, and the T_1/2_ was approximately 2.41 ± 0.1 h (Figure 6E, upper panel) This reduction in T_1/2_ rate was accelerated in the presence of AA to 1.55 ± 0.39 h; conversely, the T_1/2_ was significantly extended by more than 6 h when the cells were pretreated with MG132 as compared with AA alone (Figure 6E, lower panel). Furthermore, immunoprecipitation revealed that polyubiquitination of Bcl-2 was clearly increased in the AA-treated group but was suppressed by NAC or CC pretreatment (Figure 6F). Taken together, AA induced apoptosis via the generation of ROS and mediated Bcl-2 degradation through the AMPK/mTOR signaling pathway in NSCLC cells.

### 2.5. AA-Mediated Apoptosis in Patient-Derived Malignant Lung Cancer Cells

We then investigated the potential anticancer activity of AA on clinical specimens. The apoptosis-inducing effect was assessed in patient-derived lung cancer cells, whose genetic background was close to that of the actual human tumors, and that thereby provided an in vitro platform for the prediction of clinical response [28]. The proportion of apoptotic cells with chromatin condensation, DNA fragmentation, and apoptotic body formation was gradually increased in response to AA treatment, with a higher potency than that observed with cisplatin treatment (Figure 7A). Western blotting confirmed the molecular mechanism whereby the level of p-AMPK level was increased and those of p-mTOR and Bcl-2 were extensively decreased in the presence of AA treatment, whereas the level of their parental form remains unchanged (Figure 7B). In addition, AA exhibited a strong reduction in tumor spheroid size (Figure 7C). Thus, AA has a promising anticancer activity not only in in vitro 2D human lung cancer cells but also in an in vitro 3D patient-derived lung cancer tumorigenesis.

## 3. Discussion

In this study, we found that AA, a steroidal saponin from *A. letreae*, exhibited a potent apoptosis-inducing effect in NSCLC cells. Anticancer activities of AA have been previously reported [24], and this study also provides that AA has greater toxicity to the lung cancer cell lines than normal bronchial epithelial BEAS-2B cells, of which the selective index (SI) values were approximately 1.6–2.6-folds (Appendix A). Furthermore, we revealed the first evidence of the underlying molecular mechanism regarding the apoptosis-inducing effect. This activity was associated with an increase in ROS accumulation, leading to AMPK activation and subsequently mTOR inhibition. Consequently, the level of Bcl-2 was decreased through proteasomal degradation in a ROS/AMPK/mTOR signaling-dependent manner. Steroidal saponins, such as N45, gracillin, and dioscin, have been reported to be able to increase intracellular ROS and mediate apoptosis [29,30,31,32]. Steroidal saponins have been shown to suppress the activity of superoxide dismutase, a cellular antioxidant enzyme, and increase levels of malondialdehyde, a lipid peroxidation product, resulting in the elevation of oxidative stress [31]. Furthermore, levels of peroxiredoxin 1 and 6, which are ubiquitous thiol antioxidants, were dramatically downregulated in response to treatment with steroidal saponin, contributing to an increased ROS level and mediation of apoptosis [32]. This suggests that AA might participate in these mechanisms by interfering in the balance of intracellular pro- and antioxidant systems.

ROS has been reported to play a crucial mechanism in chemotherapy/radiation eradicating cancer cell death [33]. Cancer cells display a higher level of ROS generation that stimulates survival signaling to serve their rapid growth and proliferation; however, excessive ROS levels are able to induce cancer cell death through several pathways, including AMPK [34]. AMPK is a serine/threonine kinase known as a metabolic stress sensor that regulates various cellular activities [16,17,18]. AMPK is activated by a variety of factors, including metabolic stress, AMP, AMP mimetics, AMPK activators, and ROS, leading to its phosphorylation at threonine 172 in the α subunit and subsequently allosteric activation [35]. The suppression of mTOR is a major downstream function of AMPK in regulating energy homeostasis [36]. AMPK activation leads to inhibition of mTOR phosphorylation at serine 792 and promotes the formation of the TSC1/2 complex, which facilitates apoptosis in cancers [22,37,38]. Since mTOR plays an important role in tumor progression, AMPK-induced inactivation is a promising therapeutic strategy in cancer treatment. As our study shows, AA treatment enhanced ROS accumulation activates AMPK-suppressing mTOR and thereby induced NSCLC cell apoptosis; a phenomenon that was abolished in the presence of both the antioxidant agent NAC and an AMPK inhibitor (Figure 4C and Figure 5).

Anti-apoptotic Bcl-2, a member of the Bcl-2 family, plays a key role in regulating the mitochondrial apoptosis pathway [39]. Bcl-2 inhibits apoptosis by interfering with the oligomerization of Bax and Bak, thereby preserving the mitochondrial membrane integrity and preventing the release of cytochrome C. Post-translational modifications, such as phosphorylation, proteolytic cleavage, ubiquitination, and proteasomal degradation, can influence the overall amount and function of Bcl-2 [40,41]. Phosphorylation of Bcl-2 at serine 70 is reported to be required for polyubiquitination prior to Bcl-2 degradation via a proteasomal mechanism [27,42,43]. Dephosphorylation of Bcl-2 at serine 70 by protein phosphatase 2A (PP2A) has been shown to prevent Bcl-2 degradation, whereas knockdown of PP2A triggered Bcl-2 degradation, and hence reduced the level of Bcl-2 [27]. Consistent with this, p-Bcl-2 (Ser 70) and Bcl-2 polyubiquitination in this study appeared to increase in response to AA treatment, whereas the decrease in the level of Bcl-2 induced by AA treatment was rescued in the presence of a proteasomal inhibitor (Figure 6).

Activation of AMPK leads to the suppression of mTOR and a subsequent decrease in the level of Bcl-2. Intermittent hypoxia was shown to mediate an increase in the levels of p-AMPK and a decrease in those of p-mTOR and Bcl-2, leading to apoptosis; these effects were then reversed by an AMPK inhibitor [44]. In addition, mTOR inhibition by rapamycin contributed to NSCLC cell apoptosis by decreasing levels of Bcl-2 [45]. Rapamycin treatment also resulted in increasing polyubiquitination and degradation of several growth-regulating proteins by enhancing K-48-linked ubiquitination, which participated in proteasomal degradation [46]. We hypothesized whether the ROS/AMPK/mTOR signaling pathway and Bcl-2 degradation were relevant in the AA-inducing apoptosis mechanism. The results of this study demonstrated that pretreatment with NAC and an AMPK inhibitor decreased polyubiquitination of Bcl-2 (Figure 6F) and p-Bcl-2, compared with that in the AA-treated group (Figure 6C,D), indicating AA-induced Bcl-2 degradation through ROS/AMPK/mTOR signaling. Interestingly, ROS generation has a potent role in Bcl-2 degradation [41,47,48], and from our data, the antioxidant NAC exhibited a strong suppressive effect on Bcl-2 polyubiquitination, suggesting that ROS generation mediated by AA may directly induce Bcl-2 degradation (Figure 6F).

Importantly, the AMPK/mTOR signaling pathway is known to participate in both apoptosis and autophagic cell death [49]. There is a crosstalk pathway between these modes of cell death, although the signaling interplay between them is not completely understood. Apoptosis has been reported to be triggered because of AMPK-mediating autophagy, where autophagic inhibition contributed to apoptosis resistance [50,51]. To the best of our knowledge, autophagy is an initial response to metabolic imbalance, and prolonged excessive stress can exert apoptosis through the crosstalk mechanism. In addition, apoptosis and autophagy can occur independently, and the determination of the mode of cell death is possibly dependent on stimuli under specific circumstances. However, both apoptosis and autophagic cell death display promising mechanisms for cancer treatment. Further study on the effect of AA on autophagy is required to fully understand and support the potential application of this compound for lung cancer therapy.

In addition, the apoptosis-inducing effect of AA and its molecular mechanism were confirmed in patient-derived lung cancer cells. The experiment in patient-derived cancer cells is widely accepted as an alternative approach in cancer research that minimizes animal use and provides precise molecular mechanisms mimicking humans and corresponding to the clinical response [52,53,54,55]. The in vitro 3D-tumorigenesis model provided an adequate cancer microenvironment, in which the cancer spheroid grown on matrix-like substance displays is ultimately functional of the cells and approximately relevant to the cancer microenvironment context [56,57,58]. This model has been reported to provide a powerful tool for the investigation of cell growth and the pharmacological effect of drugs or compounds against cancers [56,57,58,59]. Using a combination of both models, our results demonstrated that AA strongly suppressed in vitro 3D-tumorigenesis, showing that there is potential in drug research and development of this compound for further clinical application.

## 4. Materials and Methods

### 4.1. Chemicals and Reagents

AA was isolated and identified as previously described (Figure 1A) [25]. AA was dissolved in dimethyl sulfoxide (DMSO) to prepare a stock solution, which was further diluted in media to the desired working concentration. DMSO was used at ≤0.1% concentration, which was non-toxic to all cells tested.

3-(4,5-Dimethylthiazol-2-yl)-2,5-diphenyltetrazolium bromide (MTT) was purchased from Invitrogen (Carlsbad, CA, USA). Hoechst 33342 and compound C (CC, Dorsomorphin) were obtained from Sigma Chemical (St. Louis, MO, USA). Annexin-V-FITC and propidium iodide (PI) were purchased from Immuno Tools (Friesoythe, Germany). MG132 was obtained from Merck Millipore (Billerica, MA, USA). Primary and secondary antibodies included: rabbit anti-Bcl-2, rabbit anti-poly-ADP-ribose polymerase (PARP), rabbit anti-caspase 3, rabbit anti-AMPK, rabbit anti-mTOR, rabbit anti-Bax, mouse anti-tubulin, anti-rabbit and anti-mouse IgG HRP-linked (Cell Signaling Technology, Beverly, MA, USA). Mouse anti-GADPH was obtained from Santa Cruz Biotechnology, Inc. (Dallas, TX, USA).

### 4.2. Ethical Approval

Patient-derived malignant cancer cells were obtained from pleural effusions from patients with recurrent or advanced-stage NSCLC. The study was approved by the Ethics Committee of the Faculty of Medicine, Chulalongkorn University, Bangkok, Thailand (IRB 365/62), and informed consent was obtained from all participants. This study was performed in accordance with the principles of the World Medical Association Declaration of Helsinki.

### 4.3. Cell Culture

NSCLC cell lines A549 (CCL-185), H460 (HTB-177), and H23 (CRL5800) were obtained from the American Type Culture Collection (ATCC; Manassas, VA, USA). The patient-derived cancer cell lines (ELC16 and ELC17) were separated from pleural effusions of recurrent or patients with advanced-stage NSCLC who had been diagnosed at the King Chulalongkorn Memorial Hospital. A549 cells were cultured in DMEM, and H460, H23 cells, and patient-derived cancer cells were cultured in RPMI-1640; both media were supplemented with 10% fetal bovine serum albumin, 2 mM L-glutamine, and 100 U/mL penicillin–streptomycin. All media and supplements were purchased from Invitrogen (Carlsbad, CA, USA). Cells were maintained in 5% CO_2_ at 37 °C.

### 4.4. Cell Viability Assay

Cytotoxicity was evaluated using the MTT assay. All tested cell lines were seeded at 8 × 10^3^ cells/well into 96-well plates. After attachment, cells were treated with various concentrations of AA for 24 h, and 10 μL of MTT (5 mg/mL) was then added to each well. After incubation for 3–4 h, 100 μL of DMSO was added to dissolve the formazan, and the optical density was measured using a microplate reader at 570 nm.

### 4.5. Apoptosis Evaluation

Apoptotic cells were assessed using Hoechst 33342 and Annexin-V-FITC/PI staining. For Hoechst 33342 staining, 8 × 10^3^ cells/well were plated into 96-well plates and incubated for 24 h. Cells were treated with 25 and 50 μM AA for 24 h followed by incubation with 1 μL Hoechst 33342 (1 mg/mL) at 37 °C for 30 min in the dark. Five random fields were selected for imaging and scoring apoptotic cells using a fluorescence microscope (Nikon Inverted Microscope Eclipse Ti-U Ti-U/B, Melville, NY, USA) at 20× magnification.

For annexin-V-FITC/PI staining, 1.3 × 10^5^ cells were seeded into 6-well plates for 24 h. Cells were treated with 25 and 50 μM AA for 24 h. Cells were harvested by centrifugation at 1000× *g* and washed with phosphate-buffered solution (PBS). Cells were incubated with 5 μL Annexin-V (25 μg/mL) and 3 μL PI (50 μg/mL) at room temperature in the dark for 15 min, and then 400 μL binding buffer was added. Apoptotic cells were evaluated using an EPICS-XL flow cytometer (Beckman Coulter, Brea, CA, USA).

### 4.6. Measurement of Reactive Oxygen Species (ROS)

Intracellular ROS levels were evaluated using a 2′,7′-dichlorofluorescein diacetate (DCFH_2_-DA) assay. Cells were plated at 8 × 10^3^ cells/well into a 96-well plate for 24 h. Cells were preincubated with DCFH_2_-DA (100 μM) for 30 min, before treatment with 25 and 50 μM AA for 3 h. Cells were washed with cold PBS, and fluorescence intensity was assessed using a microplate reader with excitation and emission at 485 and 535 nm, respectively (Anthos, Durham, NC, USA).

### 4.7. Western Blot Analysis

A total of 10^6^ cells/dish were plated onto a 60-mm dish for 24 h and were then treated with 25 and 50 μM AA for 24 h. Cells were then lysed with TMEN lysis buffer (20 mM Tris-HCl pH 7.5), 1 mM MgCl_2_, 150 mM NaCl, 20 mM NaF, 1% octylphenoxypolyethoxyethanol, 0.1 mM phenylmethylsulfonyl fluoride, 0.5% sodium deoxycholate, and cOmplete^TM^ Protease inhibitor Cocktail (Sigma-Aldrich, St. Louis, MO, USA). The supernatant was collected via centrifugation at 12,000× *g* at 4 °C for 15 min. Protein content was measured using a BSA protein assay kit (Thermo Fisher Scientific Inc., Waltham, MA, USA). An equal amount of total protein was separated using SDS-polyacrylamide gel electrophoresis and then transferred to polyvinyl difluoride membranes. The membranes were then blocked in 5% skim milk in TBS with 0.075% tween for 1 h and incubated with specific primary antibodies at 4 °C overnight. Membranes were washed three times with TBS-T and then incubated with a secondary antibody for 2 h at room temperature. Protein expression was visualized using a chemiluminescent HRP substrate (Millipore, Burlington, MA, USA). Protein bands were evaluated using ImageJ software version 1.53 (NIH, Bethesda, MD, USA).

### 4.8. Immunoprecipitation

A total of 1 × 10^6^ cells/dish were seeded onto a 60-mm dish for 24 h. Cells were pretreated with 10 μM MG132 proteasomal inhibitor for 1 h and then treated with AA in the presence or absence of 1 mM *N*-acetylcysteine (NAC) or 10 μM of compound C (CC) for 4 h. Cells were lysed using TMEN lysis buffer and placed on ice for 45 min. The supernatant was collected by centrifugation at 20,000× *g* at 4 °C for 20 min and incubated with protein G-conjugated Sepharose beads (GE Healthcare, Port Washington, NY, USA) at 4 °C for 1 h. After centrifugation at 2000× *g* at 4 °C for 3 min, the supernatant was incubated with an anti-Bcl-2 antibody at 4 °C overnight. Protein G-conjugated Sepharose beads were added and incubated further at 4 °C for 1 h and washed with lysis buffer. Protein complexes were separated by heating at 96 °C for 10 min. Polyubiquitination was investigated using Western blot analysis with an antibody to ubiquitin.

### 4.9. In Vitro Three-Dimensional (3D) Tumor Spheroid Formation Assay

Cell suspension at a number of 5 × 10^3^ cells was cultured onto 96-well plates coated with 4% Matrigel Matrigel^TM^ (BD Biosciences, Franklin Lakes, NJ, USA). After spheroids were formed, they were treated with or without AA for 5 days, in which each tumor spheroid was imaged by a Meiji Techno TC5100 inverted microscope (Saitama, Japan) with a Tuscan camera and TCapture software version 4.3.0.605 (Fujian, China). The spheroid size was analyzed by ImageJ software (NIH) and calculated as a percentage.

### 4.10. Statistical Analysis

Data are presented as the mean ± SEM from at least three independent experiments. The Student’s *t*-test and one-way ANOVA with Tukey’s multiple comparison test were applied to evaluate statistical significance between two groups and between multiple groups, respectively. A *p*-value < 0.05 was considered statistically significant.

## 5. Conclusions

This study reported an apoptosis-inducing effect of AA and the associated underlying molecular mechanism (Figure 8). AA treatment increased the level of intracellular ROS, resulting in activation of AMPK that subsequently suppressed the mTOR signaling pathway. AA treatment decreased levels of Bcl-2 by mediating its proteasomal degradation via the ROS/AMPK/mTOR axis, thereby causing an imbalance of pro- and antiapoptotic proteins and leading to apoptosis. Since lung cancer exhibits aggressive malignant tumors with a high mortality rate, this study shows that AA is a promising compound for the treatment of lung cancer and can be used in the support of anticancer drug research and development to improve clinical outcomes in the future.

## Figures and Tables

**Figure 1 ijms-23-09258-f001:**
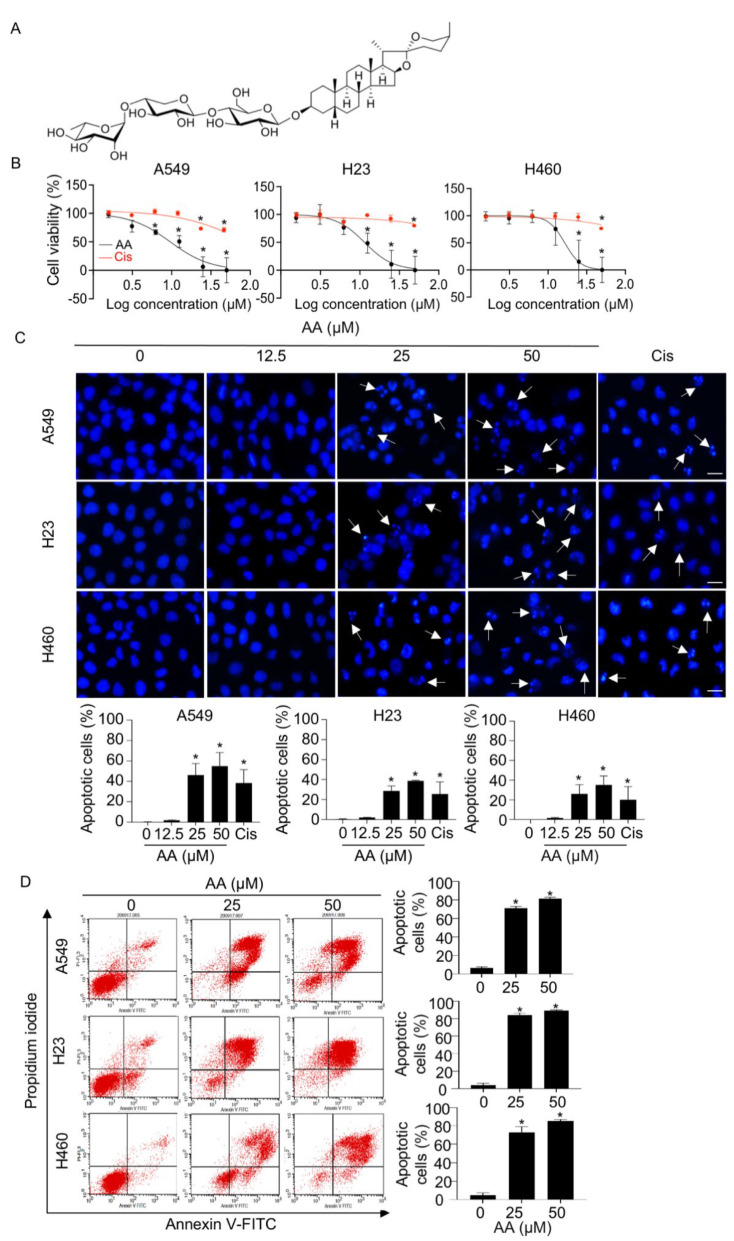
Apoptosis-inducing effect of aspiletrein A (AA) in lung cancer cells. (**A**) Chemical structure of AA. (**B**) Lung cancer A549, H23, and H460 cells were treated with either AA (0–50 μM) or cisplatin (Cis, 0–50 μM) for 24 h. Cell viability was determined via MTT assay and presented as a percentage. (**C**) Lung cancer cells were treated with AA (0–50 μM) or cisplatin (50 μM) for 24 h. Nuclei are blue in which apoptosis nuclei are indicated by the arrow as detected by H33342 staining and imaged using a fluorescence microscope. Apoptotic cells were analyzed from at least five random fields and presented as a percentage. Scale bar is 10 μm. (**D**) Lung cancer cells were treated with AA (0–50 μM) for 24 h. Apoptotic cells were examined with annexin-V-FITC/propidium iodide staining and fluorescent intensity was analyzed using flow cytometry. Red dot represented each cell that stained by annexin-V-FITC and propidium iodide. The population of annexin-V-FITC positive cells was calculated and presented as a percentage of apoptotic cells. Data are mean ± SEM (*n* = 4). * *p* < 0.05 vs. control cells.

**Figure 2 ijms-23-09258-f002:**
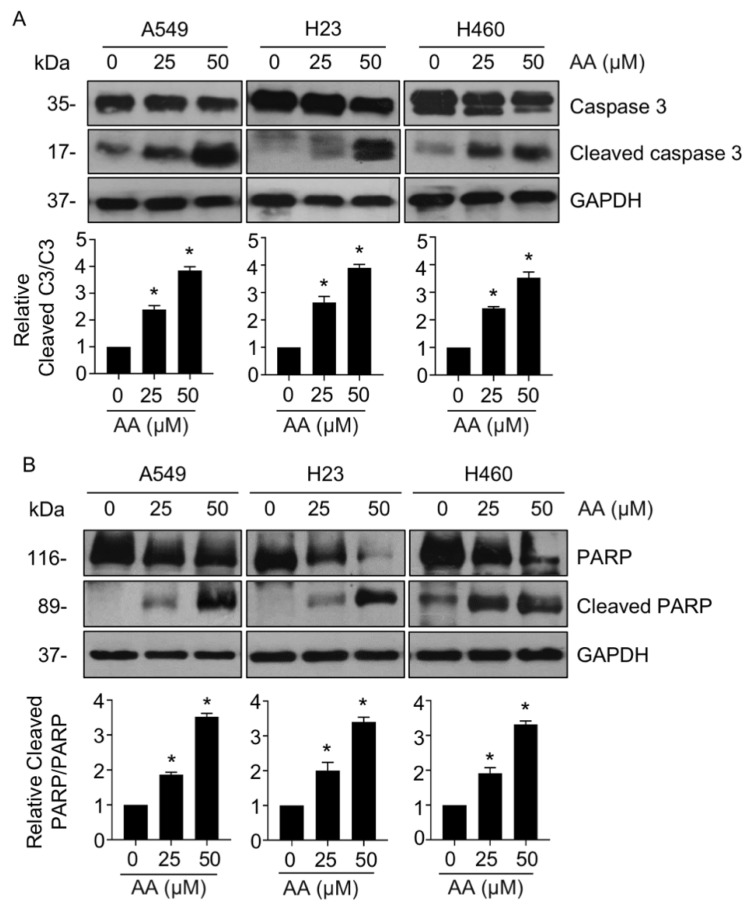
Aspiletrein A (AA) mediated cleavage of caspase-3 and poly-ADP-ribose polymerase (PARP). Lung cancer cells were treated with AA (0–50 μM) for 24 h. Protein levels of (**A**) caspase 3/cleaved caspase 3 and (**B**) PARP/cleaved PARP were analyzed using Western blotting. Blots were re-probed with anti-GAPDH to confirm equal loading. Protein level was quantified and presented as a relative value to the control. Data are mean ± SEM (*n* = 4). * *p* < 0.05 vs. control cells.

**Figure 3 ijms-23-09258-f003:**
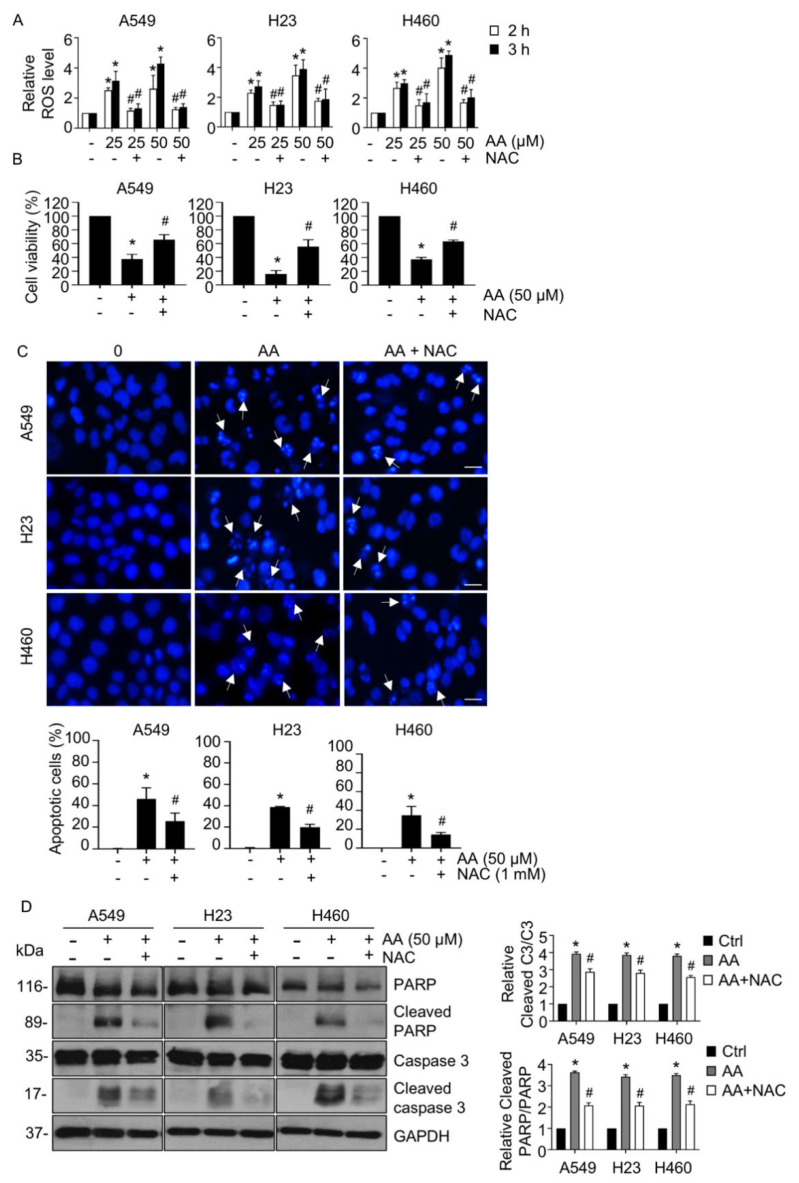
Aspiletrein A (AA) induced reactive oxygen species (ROS) generation. (**A**) Lung cancer cells were pretreated with N-acetylcysteine (NAC, 1 mM) for 30 min in the presence or absence of AA (0–50 μM) for 2 and 3 h. ROS level was assessed by DCFH_2_-DA staining and presented as a relative value to that of the control. (**B**) Lung cancer cells were pretreated with NAC (1 mM) for 30 min in the presence or absence of AA (50 μM) for 24 h. Cell viability was determined using the MTT assay. (**C**) Apoptosis nuclei were evaluated using H33342 staining and imaged under a fluorescence microscope. Nuclei are blue in which apoptotic nuclei are indicated by the arrow and analyzed from at least five random fields and presented as a percentage. Scale bar is 10 μm. (**D**) The level of caspase 3, cleaved caspase 3, PARP, and cleaved PARP were analyzed via Western blotting. Blots were re-probed with anti-GAPDH to confirm equal loading. Protein level was quantified and presented as a relative value to the control. Data are mean ± SEM (*n* = 4). * *p* < 0.05 vs. control cells, ^#^ *p* < 0.05 vs. AA-treated cells.

**Figure 4 ijms-23-09258-f004:**
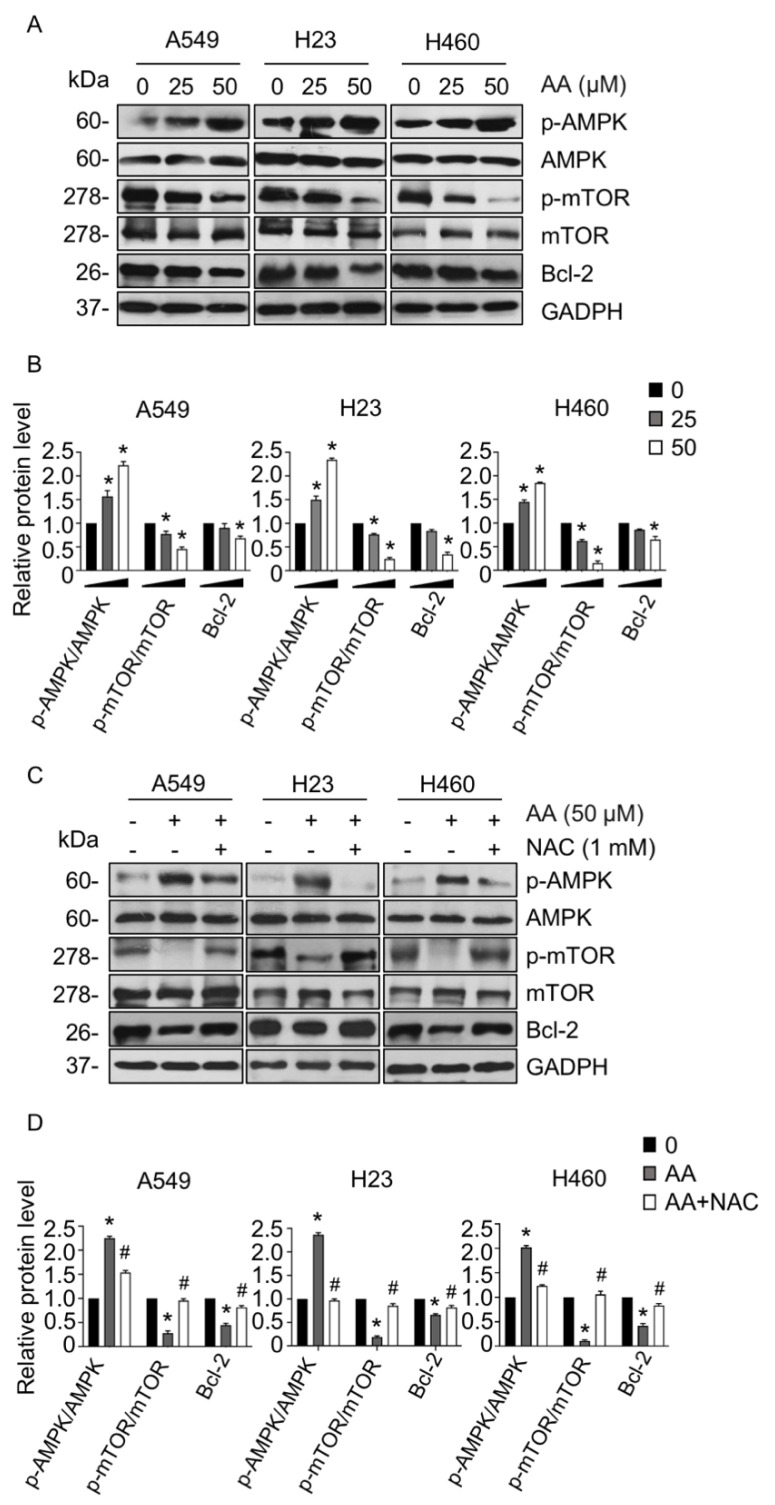
Aspiletrein A (AA) induced apoptosis via the AMPK/mTOR/Bcl-2 signaling in a ROS-dependent mechanism. (**A**) Lung cancer cells were treated with AA (0–50 μM) for 24 h. The level of p-AMPK, AMPK, p-mTOR, mTOR, and Bcl-2 were analyzed via Western blotting. Blots were re-probed with anti-GAPDH to confirm equal loading. (**B**) Protein level was quantified and presented as a relative value to the control. Data are mean ± SEM (*n* = 4). * *p* < 0.05 vs. control cells. (**C**) Lung cancer cells were pretreated with NAC (1 mM) for 30 min in the presence or absence of AA (50 μM) for 24 h. The level of p-AMPK, AMPK, p-mTOR, mTOR, and Bcl-2 were analyzed via Western blotting. Blots were re-probed with anti-GAPDH to confirm equal loading. (**D**) Protein level was quantified and presented as a relative value to that of the control. Data are mean ± SEM (*n* = 4). * *p* < 0.05 vs. control cells, ^#^ *p* < 0.05 vs. AA-treated cells.

**Figure 5 ijms-23-09258-f005:**
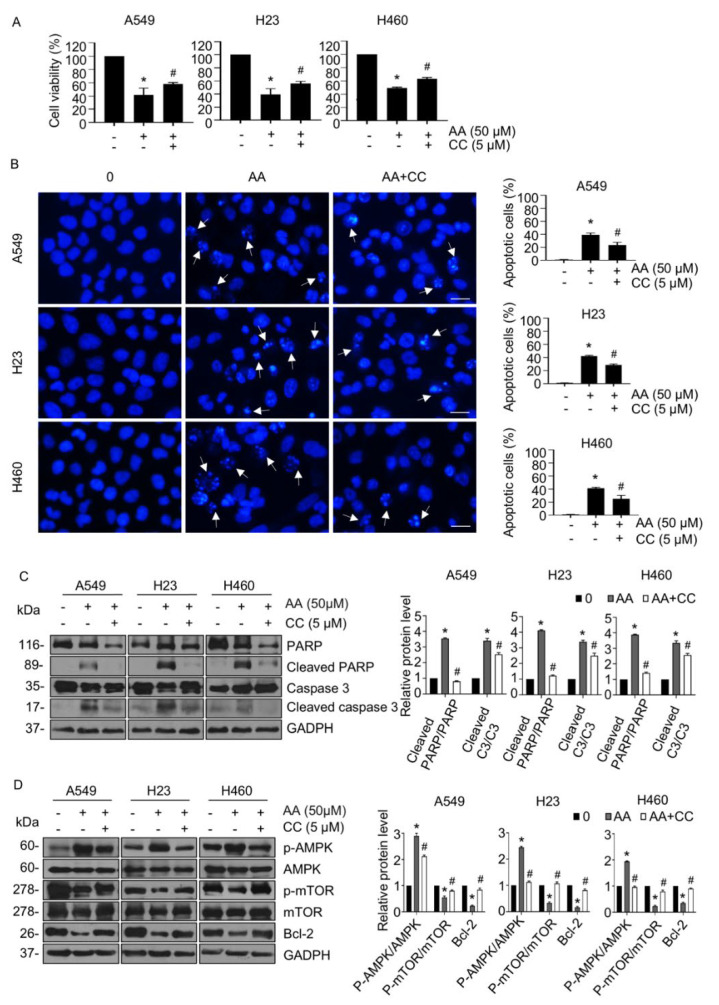
AMPK inhibitor attenuated aspiletrein A (AA)-induced apoptosis via AMPK/mTOR/Bcl-2 signaling. (**A**) Lung cancer cells were pretreated with compound C (CC, 5 μM) for 30 min in the presence or absence of AA (50 μM) for 24 h. Cell viability was determined using the MTT assay. (**B**) Apoptosis nuclei were evaluated using H33342 staining and imaged with a fluorescence microscope. Nuclei are blue in which apoptosis nuclei are indicated by the arrow and analyzed from at least five random fields and presented as a percentage. Scale bar is 10 μm. (**C**) The level of caspase 3, cleaved caspase 3, PARP, and cleaved PARP were analyzed via Western blotting. (**D**) The level of p-AMPK, AMPK, p-mTOR, mTOR, and Bcl-2 were analyzed via Western blotting. Blots were re-probed with anti-GAPDH to confirm equal loading. Protein level was quantified and presented as relative value to the control. Data are mean ± SEM (*n* = 4). * *p* < 0.05 vs. control cells, ^#^ *p* < 0.05 vs. AA-treated cells.

**Figure 6 ijms-23-09258-f006:**
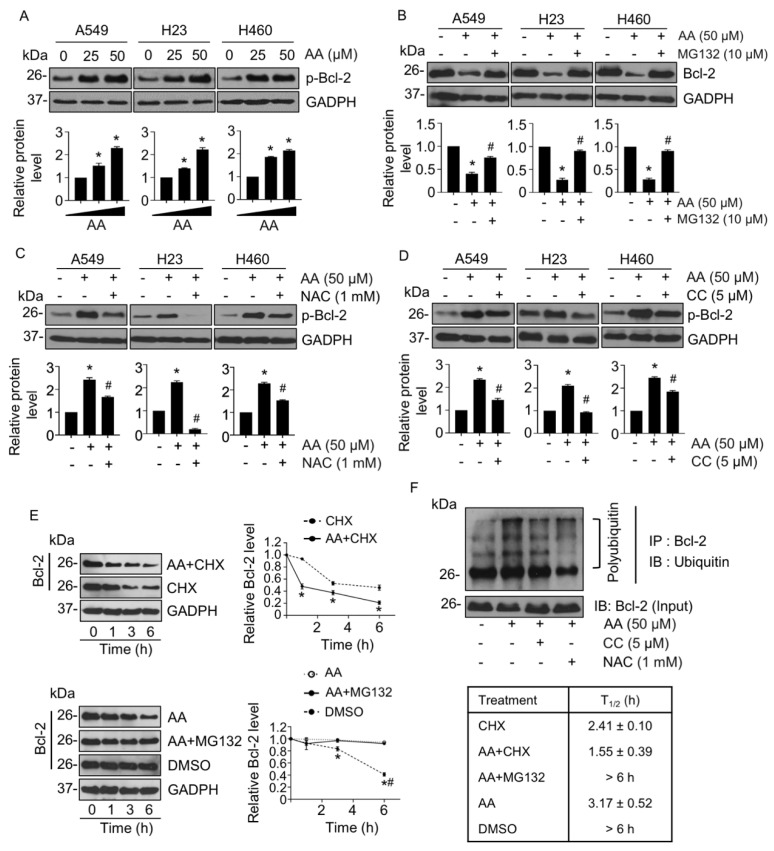
Aspiletrein A (AA) mediated Bcl-2 degradation via a ubiquitin-proteasomal mechanism. (**A**) Lung cancer cells were treated with AA (0–50 μM) for 24 h. The level of p-Bcl-2 was analyzed via Western blotting (**B**) Lung cancer cells were pretreated with MG132 (10 μM) for 1 h in the presence or absence of AA (50 μM) for 24 h. The level of Bcl-2 was analyzed via Western blotting analysis. Lung cancer cells were pretreated with (**C**) N-acetylcysteine (NAC, 1 mM) or (**D**) compound C (CC, 5 μM) in the presence or absence of AA (50 μM) for 24 h. The level of p-Bcl-2 was analyzed via Western blotting. Blots were re-probed with anti-GAPDH to confirm equal loading. Protein level was quantified and presented as relative value to the control. Data are mean ± SEM (*n* = 4). * *p* < 0.05 vs. control cells, ^#^ *p* < 0.05 vs. AA-treated cells. (**E**) Lung cancer A549 cells were pretreated with either cycloheximide (CHX, 10 μg/mL) or MG132 (10 μM) for 1 h prior to incubation with AA (50 μM) or AA alone for 0–6 h. The level of Bcl-2 was analyzed via Western blotting. Protein half-life (T_1/2_) was analyzed and presented on the right. Blots were re-probed with anti-GAPDH to confirm equal loading. Protein level was quantified and presented a relative value to that of the control. Data are mean ± SEM (*n* = 4). * *p* < 0.05 vs. CHX or DMSO treated cells, ^#^ *p* < 0.05 vs. combination of MG132- and AA-treated cells. (**F**) Lung cancer A549 cells were pretreated with or without CC (5 μM) or NAC (1 mM), prior to AA (50 μM) for 3 h. Polyubiquitination of Bcl-2 was analyzed using immunoprecipitation with an antibody to Bcl-2 and immunoblotting with an antibody to ubiquitin. Represented blots from three independent experiments are shown.

**Figure 7 ijms-23-09258-f007:**
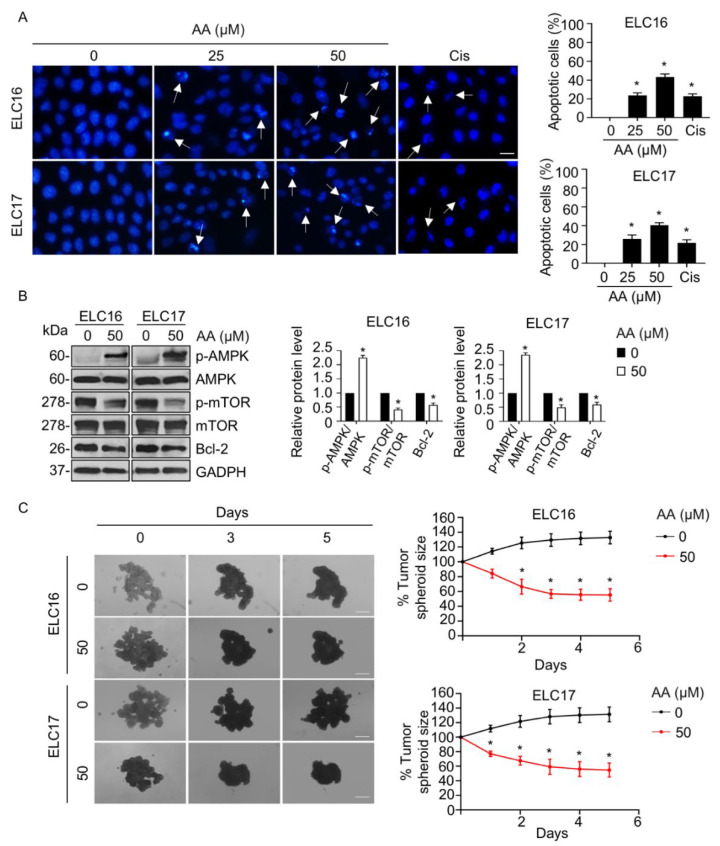
Aspiletrein A (AA) induced patient-derived lung cancer cell apoptosis. (**A**) Patient-derived lung cancer (ELC16 and ELC17) cells were treated with AA (0−50 μM) or cisplatin (50 μM) for 24 h. Nuclei are blue in which apoptosis nuclei are indicated by the arrow and detected using H33342 staining and imaged with a fluorescence microscope. Apoptotic cells were analyzed from at least five random fields and presented as a percentage. Scale bar is 10 μm. (**B**) Patient-derived lung cancer cells were treated with or without AA (50 μM) for 24 h. The level of p-AMPK, AMPK, p-mTOR, mTOR, and Bcl-2 were analyzed via Western blotting. Blots were re-probed with anti-GAPDH to confirm equal loading. Protein level was quantified and presented as a relative value to the control. (**C**) In vitro 3D tumorigenesis was performed by tumor spheroid formation assay. Tumor spheroids were treated with or without AA (50 μM) for 5 days, which were imaged every 2 days. Spheroid size was calculated as a percentage from at least 8 spheroids. Scale bar is 100 μm. All data are mean ± SEM (*n* = 4). * *p* < 0.05 vs. control cells.

**Figure 8 ijms-23-09258-f008:**
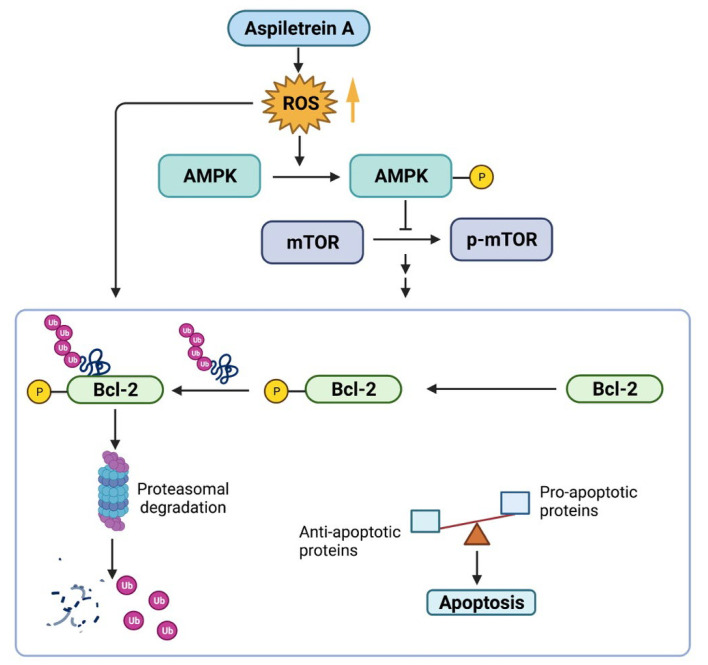
Schematic of the molecular mechanism of aspiletrein A (AA) treatment on lung cancer apoptosis. AA activates AMPK by triggering the generation of ROS, which subsequently suppresses activation of p-mTOR activation and increases levels of phosphorylated Bcl-2. This facilitates Bcl-2 ubiquitination and degradation through a proteasomal mechanism. ROS mediated by AA also directly participated in the Bcl-2 degradation. As a result, antiapoptotic Bcl-2 was downregulated, causing an imbalance of cell survival and cell death signals and thereby favoring cell apoptosis.

## Data Availability

All data supporting the findings of this study are available within the article and its Appendix A.

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
