# Peer review of "Aspiletrein A Induces Apoptosis Cell Death via Increasing Reactive Oxygen Species Generation and AMPK Activation in Non-Small-Cell Lung Cancer Cells"

_ijms, 2022, doi:10.3390/ijms23169258_

Round 1
Reviewer 1 Report
This paper is dealing with the antitumor activity of Aspiletrein. Authors convincingly demonstrate that Aspiletrin A (AA) is able to kill non small cell lung cancer cells..
The paper is well organized, methods are adequate, most of the results are clearly presented.
Major concern: Authors do not show or cite results about the selectivity of AA compound. In the absence of those data AA is only another toxic compound, even though the toxicity is very nicely analyzed.
Minor comments
30 "Consistency" did author mean consistently?
44 "survival rate dramatically decreases ... at advanced or metastatic stage" - unnecessary sentence. It is true for most of malignancies.
51 "abnormality of apoptotic cell death" please phrase it differently
79 "cytotoxicity effect" correctly: cytotoxic effect
153-155 "....indicating tha ROS generation at least participated ..." data about the ROS generation (Fig 3A do not support this conclusion".
Author Response
Reviewer #1
This paper is dealing with the antitumor activity of Aspiletrein. Authors convincingly demonstrate that Aspiletrin A (AA) is able to kill non small cell lung cancer cells..
The paper is well organized, methods are adequate, most of the results are clearly presented.
Response: We are grateful for the reviewer’s very helpful comments. We have amended the manuscript according to the reviewer’s suggestion carefully. Our point-by-point responses are below. The changes in revised MS are made in the red color. We are appreciated for further revision if some points remained required for more clarification.
1. Major concern: Authors do not show or cite results about the selectivity of AA compound. In the absence of those data, AA is only another toxic compound, even though the toxicity is very nicely analyzed.
Response: We would like to thank the reviewer for this important concern. We have mentioned and included the citation that AA has higher selectivity to lung cancer cells which has less toxic to the normal lung epithelial cells in the Introduction (line 69-70) and Discussion part (line 265-266).
2. Minor comments
- Line 30 "Consistency" did author mean consistently?
Response: We have replaced to “Consistently” in the revised MS accordingly.
- Line 44 "survival rate dramatically decreases ... at advanced or metastatic stage" - unnecessary sentence. It is true for most of malignancies.
Response: We have removed this sentence in the revised MS.
- Line 51 "abnormality of apoptotic cell death" please phrase it differently.
Response: We have rewrote to “The resistance to apoptotic cell death…”, accordingly.
- Line 79 "cytotoxicity effect" correctly: cytotoxic effect
Response: We have replaced to “cytotoxic” in the revised MS.
- Line 153-155 "....indicating tha ROS generation at least participated ..." data about the ROS generation (Fig 3A do not support this conclusion".
Response: We would like to thank the reviewer for raising an important point. We do agree with the reviewer that Fig 3A alone cannot support the conclusion this sentence (line 153-155). Figure 3 demonstrated that AA induced ROS generation and apoptosis cell death which were suppressed by anti-oxidant NAC pretreatment. It suggested that AA-induced apoptosis via ROS generation, which was mentioned in line 127-128. We then further investigated the molecular signaling regarding this effect, and Figure 4 showed that AA activated AMPK signaling pathway which was inhibited in the presence of NAC. Taken together, it suggested that ROS-mediated by AA might be an underlying mechanism on how the compound activated the AMPK signaling and apoptosis. However, we rewrote the sentence to make it clearer and more consistent with the results in the revised MS that AA-activating AMPK signaling and apoptosis was involved with ROS generation.
Reviewer 2 Report
The work presented in this article deals with the role of the compound Aspiletrein A (AA), isolated from the plant Apidistra letrae, in non-small cell lung cancer (NSCLC) cells. In particular, most of the work is centred on the action of AA on cellular apoptosis and the altered cell signalling pathways. The authors first describe that AA induces cell death via apoptosis in three NSCLC cell lines. Later on, the increase in ROS generation and activation of the AMPK-signalling pathway, including down-regulation of phosphorylated mTOR and Bcl-2, is shown. The authors showed that this response depends on the increased production of ROS because incubation with an antioxidant product (N-acetylcysteine) decreases cell apoptosis and AMPK activation. In addition, using inhibitors the authors showed that the decrease in Bcl-2 expression is dependent on ROS production and proteasomal degradation. Finally, the authors reproduce these results in cancer cells isolated from two NSCLC patients. The authors conclude from these data that AA induces apoptosis cell death in SNCLC cells and is a promising compound for the treatment of lung cancer.
The article is extensive, describes the induction of apoptosis by AA in NSCLC cells and search for the possible signalling mechanisms involved. The data are well presented and look sound. The conclusions are supported by the data presented and open new possibilities for NSCLC treatment. There are a few minor points where the manuscript could be improved, as follows:
1. Abstract, line 30 “Consistency” should probably be “Consistently”
2. It is difficult to distinguish any detail in the pictures presented in Figures 1C, 3C, 5B and 7A. The possibility of increasing their size should be considered.
3. Figure 3 legend, lines 133-134. I think the indication of panel B should be placed before “Lung cancer cells were pretreated…” (line 133).
4. Line 199. The word donwreregulation is repeated twice in the same sentence.
5. The sentence in lines 214-215 should be revised. Perhaps “AA-induced” might be changed to “AA induced”
6. Discussion, line 320, “indicating that” might better be “”indicating”
Author Response
Reviewer #2
The work presented in this article deals with the role of the compound Aspiletrein A (AA), isolated from the plant Apidistra letrae, in non-small cell lung cancer (NSCLC) cells. In particular, most of the work is centred on the action of AA on cellular apoptosis and the altered cell signalling pathways. The authors first describe that AA induces cell death via apoptosis in three NSCLC cell lines. Later on, the increase in ROS generation and activation of the AMPK-signalling pathway, including down-regulation of phosphorylated mTOR and Bcl-2, is shown. The authors showed that this response depends on the increased production of ROS because incubation with an antioxidant product (N-acetylcysteine) decreases cell apoptosis and AMPK activation. In addition, using inhibitors the authors showed that the decrease in Bcl-2 expression is dependent on ROS production and proteasomal degradation. Finally, the authors reproduce these results in cancer cells isolated from two NSCLC patients. The authors conclude from these data that AA induces apoptosis cell death in SNCLC cells and is a promising compound for the treatment of lung cancer.
The article is extensive, describes the induction of apoptosis by AA in NSCLC cells and search for the possible signalling mechanisms involved. The data are well presented and look sound. The conclusions are supported by the data presented and open new possibilities for NSCLC treatment. There are a few minor points where the manuscript could be improved, as follows:
Response: We are grateful for the reviewer’s very helpful comments. We have amended the manuscript according to the reviewer’s suggestion carefully. Our point-by-point responses are below. The changes in revised MS are made in the red color. We are appreciated for further revision if some points remained required for more clarification.
1. Abstract, line 30 “Consistency” should probably be “Consistently”
Response: We have replaced to “Consistently” in revised MS accordingly.
2. It is difficult to distinguish any detail in the pictures presented in Figures 1C, 3C, 5B and 7A. The possibility of increasing their size should be considered.
Response: We would like to thank the reviewer for this important concern. We have enlarged the images and included arrows indicating apoptosis cells.
3. Figure 3 legend, lines 133-134. I think the indication of panel B should be placed before “Lung cancer cells were pretreated…” (line 133).
Response: We have revised the Figure legend according to reviewer’s suggestion.
4. Line 199. The word donwreregulation is repeated twice in the same sentence.
Response: We have removed the repeated word in revised MS accordingly.
5. The sentence in lines 214-215 should be revised. Perhaps “AA-induced” might be changed to “AA induced”
Response: We have revised this point in revised MS accordingly.
6. Discussion, line 320, “indicating that” might better be “indicating”
Response: We have replaced to “indicating” in revised MS (now in line 321).

Reviewer 3 Report
Dear authors,
Congratulations on the performed hard work to prepare the manuscript.
Your findings revealed the underlying mechanism of Aspiletrein A's effects on lung cancer apoptosis, opening the possibility of additional clinical investigation and development of this compound. Research into compounds that may be potential candidates for use in the treatment of cancer patients is particularly important.
In the manuscript, data are well presented in corresponding Figures, with information about statistical significance.
The obtained results were discussed in relation to the previously published data and based on this study's novelty, which is clearly emphasized.
Kind regards,
Author Response
Reviewer #3
Congratulations on the performed hard work to prepare the manuscript.
Your findings revealed the underlying mechanism of Aspiletrein A's effects on lung cancer apoptosis, opening the possibility of additional clinical investigation and development of this compound. Research into compounds that may be potential candidates for use in the treatment of cancer patients is particularly important.
In the manuscript, data are well presented in corresponding Figures, with information about statistical significance.
The obtained results were discussed in relation to the previously published data and based on this study's novelty, which is clearly emphasized.
Response: We are grateful for the reviewer’s very helpful comments and revised MS. We have amended the manuscript according to the reviewer’s suggestion carefully. Our point-by-point responses are below. The changes in revised MS are made in the red color. We are appreciated for further revision if some points remained required for more clarification.
Round 2
Reviewer 1 Report
As a referee my major concern was about the paper:
„Authors do not show or cite results about the selectivity of AA compound. In the absence of those data, AA is only another toxic compound, even though the toxicity is very nicely analyzed.”
Authors reply: We would like to thank the reviewer for this important concern. We have mentioned and included the citation that AA has higher selectivity to lung cancer cells which has less toxic to the normal lung epithelial cells in the Introduction (line 69-70) and Discussion part (line 265-266).
Referee’s reply: Unfortunately, I have to ask for further amendments in the text.
The citation included in the 1st version of the MS: “A recent study revealed that AA could attenuate lung cancer metastasis through suppression of the protein kinase B (Akt) signaling pathway and induced cell death by downregulation of the anti-apoptotic Bcl-2 protein [24]”. This particular sentence does not mention anything about the effects of AA on the normal lung epithelial cells. The reference [24] has the title: „Antitumor activities of Aspiletrein A, a steroidal saponin from Aspidistra letreae, on non-small cell lung cancer cells.” In the abstract of the reference [24] there is not a word about the tumor-selective effect of the compound. Reading the whole paper (ref. 24) investigations on non cancer pulmonary cells were mentioned once in the Table I. and the same results are written in the results. The name of the control cells is mentioned only once (BEAS-2B) in the legend of Table 1. Authors of ref 24 (partially overlap with authors of the present MS) do not mention the culture conditions of this BEAS-2B, Therefore the most important informations upon the current study is based are not available for the readers and for the referees.
In the discussion authors again cite ref 24 “Anticancer activities of AA have been previously reported which has higher selectivity to lung cancer cells [24].
My suggestion: authors in this MS should put more emphasis on the data available on the effects of AA on normal bronchial epithelial cells. Actually the experiments performed in the first version of the MS should be done on control cell lines.
Some references suggested for the comparison between drugs selectivity index: doi: 10.7150/ijbs.34878; https://
doi.org/10.3390/plants10102193;
Author Response
Reviewer #1
As a referee my major concern was about the paper:
“Authors do not show or cite results about the selectivity of AA compound. In the absence of those data, AA is only another toxic compound, even though the toxicity is very nicely analyzed.”
Authors reply: We would like to thank the reviewer for this important concern. We have mentioned and included the citation that AA has higher selectivity to lung cancer cells which has less toxic to the normal lung epithelial cells in the Introduction (line 69-70) and Discussion part (line 265-266).
Referee’s reply: Unfortunately, I have to ask for further amendments in the text.
The citation included in the 1st version of the MS: “A recent study revealed that AA could attenuate lung cancer metastasis through suppression of the protein kinase B (Akt) signaling pathway and induced cell death by downregulation of the anti-apoptotic Bcl-2 protein [24]”. This particular sentence does not mention anything about the effects of AA on the normal lung epithelial cells. The reference [24] has the title: “Antitumor activities of Aspiletrein A, a steroidal saponin from Aspidistra letreae, on non-small cell lung cancer cells.” In the abstract of the reference [24] there is not a word about the tumor-selective effect of the compound. Reading the whole paper (ref. 24) investigations on non cancer pulmonary cells were mentioned once in the Table I. and the same results are written in the results. The name of the control cells is mentioned only once (BEAS-2B) in the legend of Table 1. Authors of ref 24 (partially overlap with authors of the present MS) do not mention the culture conditions of this BEAS-2B, Therefore the most important informations upon the current study is based are not available for the readers and for the referees.
In the discussion authors again cite ref 24 “Anticancer activities of AA have been previously reported which has higher selectivity to lung cancer cells [24].
My suggestion: authors in this MS should put more emphasis on the data available on the effects of AA on normal bronchial epithelial cells. Actually the experiments performed in the first version of the MS should be done on control cell lines.
Some references suggested for the comparison between drugs selectivity index: doi: 10.7150/ijbs.34878; https://doi.org/10.3390/plants10102193;
Response: We would like to thank the reviewer for important point and apologize for our misunderstanding in previous response. Previous study showed that even though AA has toxicity to normal lung epithelial BEAS-2B cells, the concentration of AA that causes BEAS-2B cell death was greater than those found in lung cancer cell lines (Nguyen et al., 2021). However, the selectivity index (SI), which calculated from IC50 of normal cells to IC50 of cancer cells, was approximately 1.6-2.6 folds. The greater SI value indicates the higher selectivity to cancer (Singh et al., 2019). It suggested that AA has high selectivity to lung cancer cells.
Additionally, we conducted a new set of cytotoxicity experiment in the present work. The IC50 of AA on BEAS-2B cells was 25.11 ± 4.73. The SI was analyzed as Table S1 below which is consistent with previous study. We have mentioned this point in the Discussion part of revised MS and supplementary information.
Table S1 Cytotoxicity of AA against lung cancer and normal lung epithelial cell lines.
|
Lung cancer cell lines |
IC50 (µM ± SD) |
Bronchial epithelial cell line |
IC50 (µM ± SD) |
SI |
|
A549 |
9.60 ± 2.57 |
BEAS-2B |
25.11 ± 4.73 |
2.6 |
|
H23 |
11.43 ± 3.07 |
2.19 |
||
|
H460 |
15.44 ± 3.29 |
1.62 |
References
Nguyen HM, Nguyen HT, Seephan S. et al. Antitumor Activities of Aspiletrein A, a Steroidal Saponin from Aspidistra letreae, on Non-small Cell Lung Cancer Cells. BMC Complement Med Ther. 2021;21:87.
Singh K, Gangrade A, Jana A, Mandal BB, Das NJAO. Design, synthesis, characterization, and antiproliferative activity of organoplatinum compounds bearing a 1, 2, 3-triazole ring. ACS Omega. 2019;4:835–41.
